# Characterising the Robustness of Reinforcement Learning for Continuous Control using Disturbance Injection

**Catherine R. Glossop**[*]
Department of Engineering Science
University of Toronto
`catherine.glossop@robotics.utias.utoronto.ca`

**Jacopo Panerati**
Institute for Aerospace Studies
University of Toronto
`jacopo.panerati@utoronto.ca`

**Amrit Krishnan**
Vector Institute
`amritk@vectorinstitute.ai`

**Zhaocong Yuan**
Institute for Aerospace Studies
University of Toronto
`justin.yuan@mail.utoronto.ca`

**Angela P. Schoellig**
Institute for Aerospace Studies
University of Toronto
`angela.schoellig@utoronto.ca`

## Abstract

In this study, we leverage the deliberate and systematic fault-injection capabilities of an open-source benchmark suite to perform a series of experiments on state-of-the-art deep and robust reinforcement learning algorithms. We aim to benchmark robustness in the context of continuous action spaces—crucial for deployment in robot control. We find that robustness is more prominent for action disturbances than it is for disturbances to observations and dynamics. We also observe that state-of-the-art approaches that are not explicitly designed to improve robustness perform at a level comparable to that achieved by those that are. Our study and results are intended to provide insight into the current state of safe and robust reinforcement learning and a foundation for the advancement of the field, in particular, for deployment in robotic systems.

## 1 Introduction

Reinforcement learning (RL) has become a promising approach for robotic control, showing how robotic agents can learn to perform a variety of tasks, such as trajectory tracking and goal-reaching, on several robotic systems, from robotic manipulators to self-driving vehicles [10, 22, 26]. While many of these results have been achieved in highly controlled simulated environments [13], the next wave of artificial intelligence (AI) research is now faced with the challenge to deploy these RL control approaches in the real world.

When using reinforcement learning to solve these real-world problems, safety must be paramount [27, 5, 2, 30, 12, 3]. Unsafe interaction with the environment and/or people in that environment can have very serious consequences, ranging from the destruction of the robot itself to, most importantly, harm to humans. For safety to be guaranteed, an embodied RL agent (i.e., the robot) must satisfy the constraints that define its safe behaviour (i.e., not producing actions that damage the robot, hit

---

[*]Work done during an internship at the Vector Institute for Artificial Intelligence

36th Conference on Neural Information Processing Systems (NeurIPS 2022).

obstacles or people, etc.) and be robust to variations in the environment, its dynamics, and unseen situations that can emerge in the real world.

In this article, we quantitatively study and report on the performance of a set of state-of-the-art reinforcement learning approaches in the context of continuous control. We systematically evaluate RL agents (or "controllers") on their performance (i.e., the ability to accomplish the task specified by the environment's reward signal) as well as their robustness [35, 7, 14, 16, 18], which entails a bounded form of generalisability. To do so, we used an open-source RL safety benchmarking suite [34]. First, we empirically compare the control policies produced by both traditional and robust RL agents at baseline and then when a variety of disturbances are injected into the environment.

What we observe is that both the traditional and robust RL agents are more robust to disturbances injected through the actions of the agent while disturbances injected at the level of the observations and dynamics of the agent cause much more rapid destabilisation. We also note that traditional "vanilla" agents show similar performance to the robust RL agents even when disturbances are injected, despite not being explicitly designed with this purpose in mind. By leveraging open-source simulations and implementations, we hope that this work and our insights can provide a basis for further research into safe and robust RL, especially for robot control.

## 2   Background

In RL, an agent, in our case, a robot, performs an action and receives feedback (reward) from the environment on how well it is doing at the environment's task, perceives the updated state of the environment resulting from the action taken and repeats the process, learning over time to improve the actions it takes to maximise reward collection (and this to correctly perform the task). The resulting behaviour is called the agent's policy and maps the environment's state to actions [28]. While early RL research was demonstrated in the context of grid worlds and games, in recent years, we have seen a growing interest in physics-based simulation for robot learning [8, 11, 19, 6]. For simplicity and reproducibility reasons, however, many of these simulators are still fully deterministic (and prone to be exploited by the agents).

In this study, we deliberately inject disturbances at different points of the RL learning and control interaction loop to emulate the conditions an agent might encounter in the real world. For the sake of brevity, the results reported in Sections 4 pertain to the classical cart-pole stabilisation task. In the Supplementary Material we include results for the more complex tasks of quadrotor trajectory tracking and stabilisation.

### 2.1   Injecting Disturbances in Robotic Environments

We systematically inject each of the disturbances in Figure 2 in one of three possible sites: observations, actions, and dynamics of the environment that the RL agent interacts with.

**Observation/state Disturbances**    Observation/state disturbances occur when the robot's sensors cannot perceive the exact state of the robot. This is a very common problem in robotics and is tackled with state estimation methods [1]. In the case of the cart-pole, this disturbance is four-dimensional— as is the state—and is measured in metres in the first dimension, radians in the second, metres per second in the third, and radians per second in the fourth. This disturbance is implemented by directly modifying the state observed by the system. For the quadrotor task in the Supplementary Material, observation disturbance is similarly added to the six-dimensional drone's true state.

**Action Disturbances**    Action disturbances occur when the actuation of the robot's motors is not exactly as the control output specifies, resulting in a difference between the actual and expected action. For example, action delays are often neglected or coarsely modeled in simple simulations. In the case of the cart-pole, this disturbance is a one-dimensional force (in Newtons) in the $x$-direction directly applied to the slider-to-cart joint. For the quadrotor task, action disturbances are similarly added to the UAV's commanded individual motor thrusts.

**External Dynamics Disturbances**    External dynamics disturbances are disturbances directly applied to the robot that can be thought of as environmental factors such as wind or other external forces. In the case of the cart-pole, this disturbance is two-dimensional and implemented as a tapping force (in

Newtons) applied to the top of the pole. For the quadrotor task, the dynamics disturbance is a planar wind force applied directly to the drone's centre of mass.

## 2.2 Reinforcement Learning Agents for Continuous and Robust Control

While some of the most notable results of deep RL control [15] were achieved in the context of discrete action spaces, we focus on actor-critic agents capable of dealing with the continuous actions spaces needed for embodied AI and robotics [21, 17]. Here, we summarise the agent whose results we report in Section 4. Results for additional agents are included in the Supplementary Material.

**Proximal Policy Optimisation (PPO)**   PPO [25] is a state-of-the-art policy gradient method proposed for the tasks of robot locomotion and Atari game playing. It improves upon previous policy optimisation methods such as ACER (Actor-Critic with Experience Replay) and TRPO (Trust Region Policy Optimisation) [24]. PPO reduces the complexity of implementation, sampling, and parameter tuning using a novel objective function that performs a trust-region update that is compatible with stochastic gradient descent.

**Soft Actor-Critic (SAC)**   SAC [9] is an off-policy actor-critic deep RL algorithm proposed for continuous control tasks. The algorithm merges stochastic policy optimisation and off-policy methods like DDPG (Deep Deterministic Policy Gradient). This allows it to better tackle the exploration-exploitation trade-off pervasive in all reinforcement learning problems by having the actor maximise both the reward and the entropy of the policy. This helps to increase exploration and prevent the policy from getting stuck in local optima.

**Robust Adversarial Reinforcement Learning (RARL)**   Unlike the previous two approaches, RARL [20], as well as the following approach, RAP, are designed to be robust and bridge the gap between simulated results for control and performance in the real world. To achieve this, an adversary is introduced that learns an optimal destabilisation policy and applies these destabilising forces to the agent, increasing its robustness to real disturbances.

**Robust Adversarial Reinforcement Learning with Adversarial Populations (RAP)**   RAP [33] extends RARL by introducing a population of adversaries that are sampled from and trained against. This algorithm hopes to reduce the vulnerability that previous adversarial formulations had to new adversaries by increasing the kinds of adversaries and therefore adversarial behaviours seen in training. Similar to RARL, RAP was originally evaluated on continuous control problems.

# 3   Experimental Setup

Our objective then is to train the agents in Section 2 to perform a task (e.g. cart-pole stabilisation in Section 4, quadrotor trajectory tracking in the Supplementary Material) in ideal conditions (i.e., without disturbances) and then assess the robustness of the resulting policies in environments that include injected disturbances.

Each RL agent was trained by randomising the initial state across episodes to improve performance [34] while at test/evaluation time, a unique initial state was used for fairness and consistency. The range of disturbance levels used in each experiment was selected to include (low) values, at which all or most agents still succeeded in completing the tasks, up until (high) values at which the robustness of all agents eventually fails.

In the case of the cart-pole, the goal of the controller is to stabilise the system at a pose of 0 m, or centre, in $x$ and 0 rads in $\theta$, when the pole is upright. The quadrotor is required to track a circular reference trajectory on the $x$-$z$ plane with a 0.5 m radius and an origin at (0,0,0.5). The trajectory gives a way-point at each control step and is appended in the observation for the next action.

**Evaluation Metric**   To measure the performance of the control policies, the exponentiated negated quadratic return is averaged over the length of each episode, over 25 evaluation episodes. The same metric was used for training and evaluation.

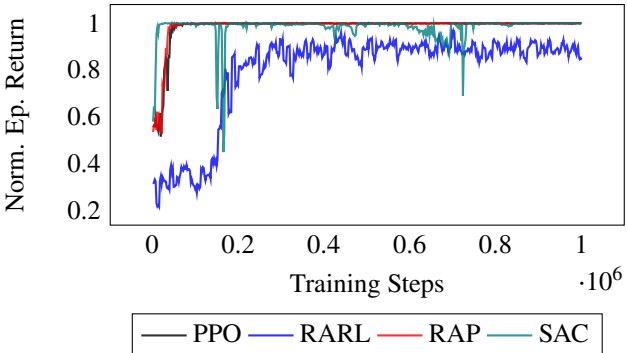

Figure 1: Training curves (# of training steps vs. returns) normalized and averaged over 10 runs in environments without disturbances for the agents in Section 2 on the cart-pole stabilisation task.

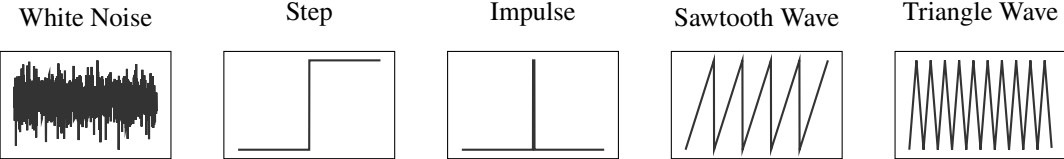

Figure 2: Appearance (in one dimension vs. time) of the disturbances injected in the experiments in Section 4: white noise, step, impulse, sawtooth, and triangle waves.

$$\text{Cost}: J_i^Q = (x_i - x_i^{goal})^T W_x (x_i - x_i^{goal}) + (u_i - u_i^{goal})^T W_u (u_i - u_i^{goal}) \quad (1)$$

$$\text{Ep. Return}: J^R = \sum_{i=0}^{L} \exp(-J_i^Q) \quad (2)$$

$$\text{Avg. Norm. Return}: J_{eval}^R = \frac{1}{N} \sum_{j=0}^{N} \frac{J_j^R}{L_j} \quad (3)$$

Equation (1) shows the task's cost computed at each episode's step $i$, where $x$ and $x^{goal}$ are the actual and goal states of the system, $u$ and $u^{goal}$ the actual and goal inputs, and $W_x$ and $W_u$ are constant weight matrices. $L$ is the total number of steps in a given episode. Equation (2) shows how to compute the return of an episode $j$ of length $L_j$ from the cost function. $L_j$ is equal (or lower) than the maximum episode duration of 250 steps. Equation (3) shows the average return for $N$ (25) evaluation runs normalised by the length of the run. This evaluation metric is the one used through Section 4.

## 4 Experiments

In Figure 1, we can look at the training results when no additional disturbances are applied, showing the reference performance of each controller at baseline. The three algorithms which reach convergence fastest were SAC, PPO, and RAP. SAC and PPO benefit from the stochastic characteristics of their updates. RARL trains more slowly which is what we expect as RARL is also learning to counteract the adversary. However, the same behaviour is not observed for the other robust approach RAP, which also converges quickly, suggesting RAP can be trained more efficiently.

### 4.1 Non-periodic Disturbances

Having trained the four agents (PPO, SAC, RARL, RAP), we want to assess the robustness of the resulting policies. In Figure 2, we introduce five types of disturbances, three non-periodic and two periodic ones that are studied in this (4.1) and the following Subsection 4.2.

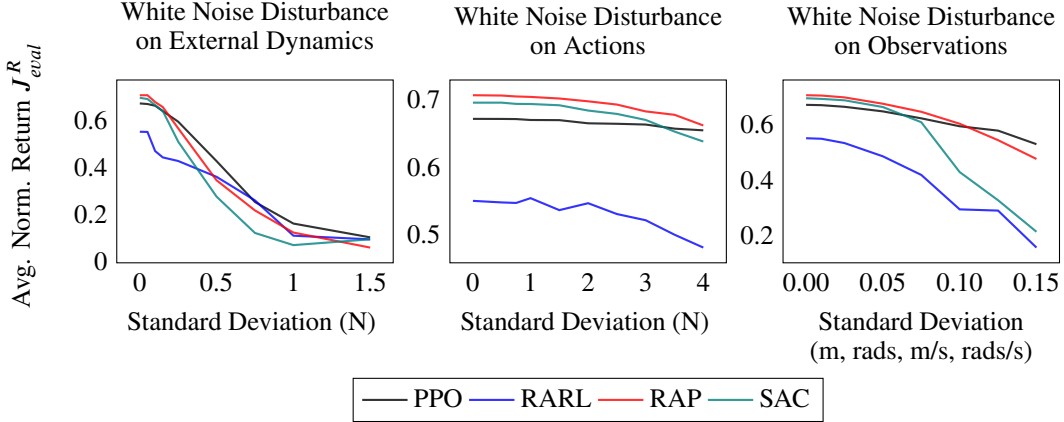

Figure 3: Average normalised return with the injection of white noise disturbances applied to (left to right) dynamics, actions, and observations on the cart-pole stabilisation task for the four RL agents.

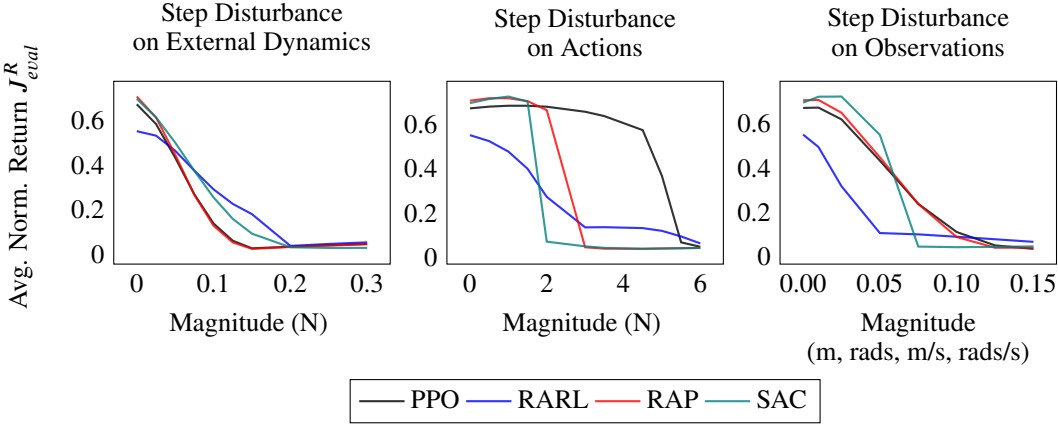

Figure 4: Average normalised return with the injection of step disturbances applied to (left to right) dynamics, actions, and observations on the cart-pole stabilisation task for the four RL agents.

**White Noise Disturbances** We first look at white noise disturbances (Figure 2a), often used to mimic the natural stochastic noise that an agent encounters in the real world. The noise is applied, from zero, at increasing values of standard deviation. In Figure 3, we see very similar low robustness across all control approaches for disturbances on external dynamics, with, as expected, a linear decrease in performance as the noise increases. However, the robust approaches, RARL and RAP, show no significant difference in performance w.r.t. PPO. Additional ablations for the robust methods are included in the Supplementary Material.

For action disturbances, RAP consistently has the highest average normalised return. For observation disturbances, PPO has the highest average normalised return at high levels of disturbances. Overall, the difference across the four approaches is small and they all demonstrate similarly good robustness when white noise is applied to observations or actions.

**Step Disturbances** Step disturbances (Figure 2b) allow us to see the system's response to a sudden and sustained change. As in all experiments, the disturbance is applied at varying levels, here representing the magnitude of the step. The step occurs two steps into the episode for all runs.

As expected, compared to white noise disturbances, step disturbances have a much greater effect and even low magnitudes result in a large decrease in performance. There are especially steep decreases

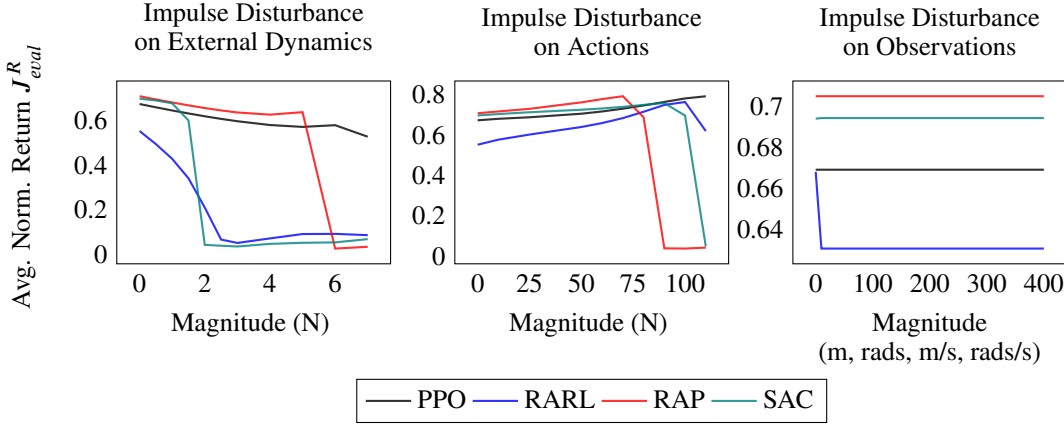

Figure 5: Average normalised return with the injection of impulse disturbances applied to (left to right) dynamics, actions, and observations on the cart-pole stabilisation task for the four RL agents.

in average normalised return when the agent can no longer stabilise the cart-pole, e.g., in the second and third plots of Figure 4.

For the step disturbance on external dynamics, there is no unique better controller, with RARL marginally outperforming the others. For actions and observations, PPO again achieves the best overall performance, yielding almost ideal average normalised return up until the step magnitude reaches, for action disturbances, as high as 5 N. SAC's performance is technically higher at low levels of disturbances but fails quickly as the magnitude of the step increases.

**Impulse Disturbances**    Impulse disturbances (Figure 2c) allow us to see the system's response to a sudden, but temporary change. Again, we look at varying levels of the impulse's magnitude to test the controllers' robustness. The width of the impulse is two steps and it is applied two steps into the run for all runs.

In the case of dynamic and action impulse disturbances, the dramatic decrease in performance seen in the previous experiment (with the step disturbances) is just as pronounced. We expected impulse disturbances to be more easily handled than step disturbances, as step responses may require the system to adapt to a new baseline whereas the impulse disturbances' change is only temporary. However, the first two plots in Figure 5 show a dramatic change in average normalised return as the sharp disturbance causes the agent to fail to stabilise. PPO is the most robust to impulse disturbances on external dynamics while RARL displays more robust performance than it did with step disturbances.

For disturbances applied to actions, SAC, PPO, and RARL are able to handle higher magnitudes of impulse disturbance than RAP. On the other hand, the short-lived impulse disturbance on observations does not significantly affect any of the control approaches, even at very high values.

### 4.2 Periodic Disturbances

Beyond episodic disturbances, we also want to explore the ability of the controllers to deal with periodic disturbances. Such disturbances further challenge the agents as they introduce long-lasting perturbations, that force them to seek robustness in an environment that never behaves as ideal.

**Saw Wave Disturbances**    A saw (or sawtooth) wave (Figure 2d) is a cyclic wave that increases linearly to a set magnitude and instantaneously drops back to a starting point before repeating the cycle. Thus, this disturbance type includes aspects of the step and impulse disturbances, yet it is applied periodically throughout the evaluation episodes.

In Figure 6, the difference in performance between the approaches is less marked (in comparison to the disturbances applied in previous experiments). For disturbances in the dynamics, there is little difference in performance (and low overall robustness) for all control approaches. RARL performs better than the other approaches at low amplitude disturbances. For action disturbances, PPO is the

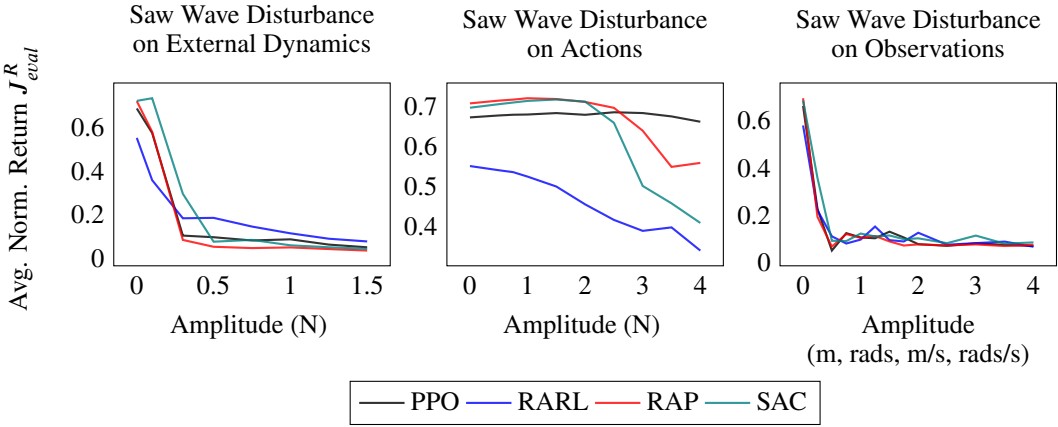

Figure 6: Average normalised return with the injection of sawtooth wave disturbances applied to (left to right) dynamics, actions, and observations on the cart-pole stabilisation task for the four RL agents.

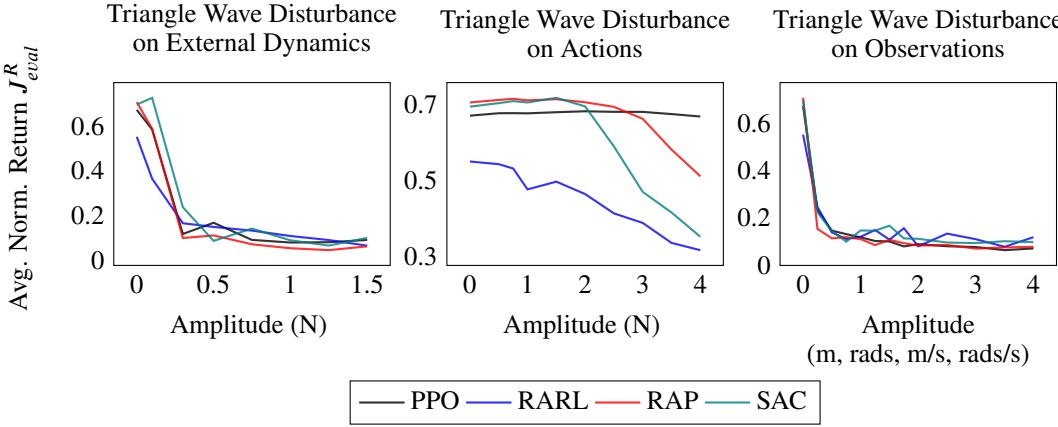

Figure 7: Average normalised return with the injection of triangle wave disturbances applied to (left to right) dynamics, actions, and observations on the cart-pole stabilisation task for the four RL agents.

agent that best preserves its average normalised return, while the other approaches, in particular SAC, display lower robustness.

When the policy behaviour of the controllers was re-played, it was evident that RAP and RARL failed more often than PPO and SAC, resulting in a lower average normalised return. When the saw wave disturbance is applied to observations, all approaches have great difficulty stabilising and the average normalised return quickly drops to zero.

**Triangle Wave Disturbance**    A triangle wave (Figure 2e) is a cyclic wave that increases linearly to a set magnitude and decreases at the same rate to a starting point before repeating. This disturbance type is very similar to the saw wave disturbance but also acts more similarly to a sinusoidal wave.

Not surprisingly, the results for triangle wave disturbances (Figure 7) are similar to those of the sawtooth wave disturbances. The triangle wave disturbance results in a slightly lower average normalised return than the sawtooth wave disturbances but the relative performance of the control approaches remains the same. SAC performs slightly worse in the case of disturbances applied to dynamics. For disturbances applied to observations, the drop in performance occurs even earlier for all controllers, showing the increased sensitivity to the triangle wave disturbance compared to the sawtooth wave.

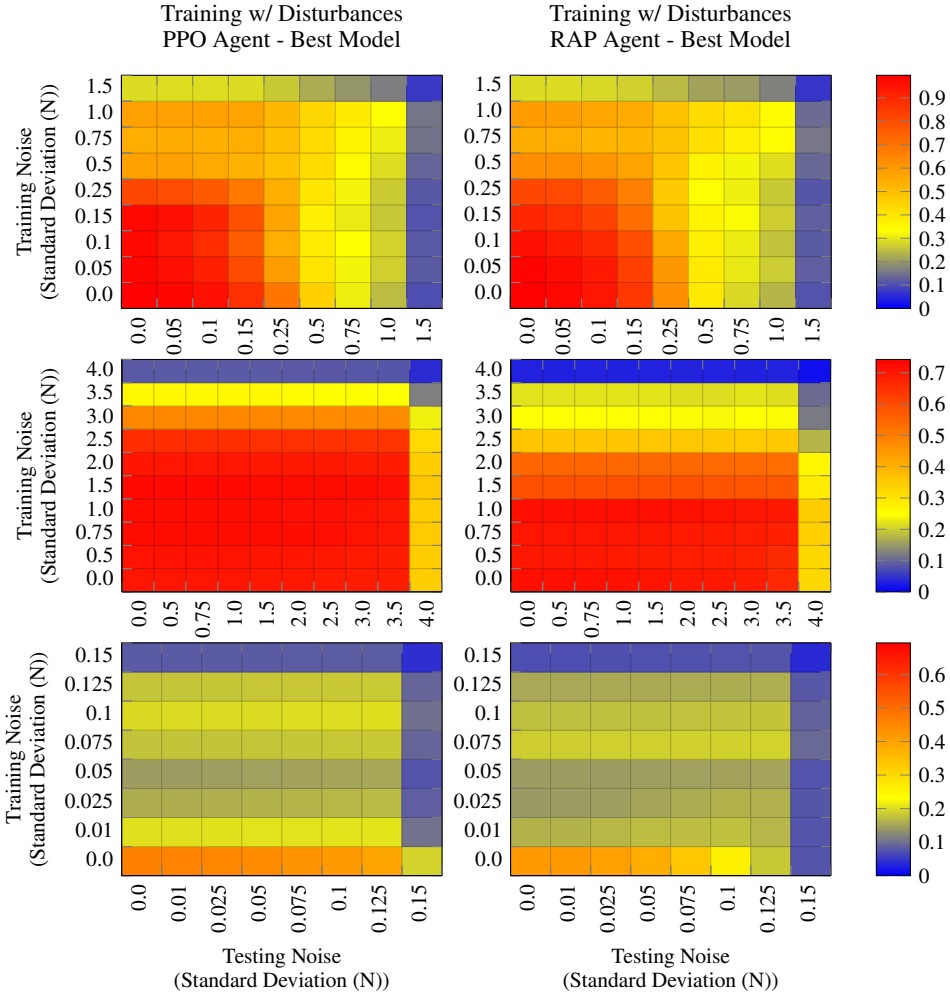

Figure 8: Heat maps of the average normalised return for PPO (left) and RAP (right) trained on (*y*-axes) and tested (*x*-axes) with varying levels of white noise on the cart-pole stabilisation task. Rows, from top to bottom, report on disturbances in dynamics, actions, and observations.

## 4.3 Training with Disturbances

In Subsections 4.1 and 4.2, no additional disturbances were introduced during training. It is natural to wonder whether including disturbances during training (and reducing the distributional shift between train and test scenario) can improve the evaluation performance of the controllers. Disturbances during training—akin to how the RARL and RAP use adversaries to increase their robustness—can potentially lead to the learning of more generalisable policies. In Figure 8, we look at two of the control approaches, PPO, the best performing vanilla RL approach, and RAP, the best performing robust approach, trained with varying levels of white noise (for 1,000,000 steps).

The evaluation/test performance with higher levels of noise is almost always still better when training with low levels of noise, and achieving the best performance when trained with no disturbances. For external dynamics disturbances, the average normalised return gradually decreases as the training noise is increased. At higher values of training noise, the performance when the levels of testing noise are also higher improves slightly, suggesting there are small improvements. This phenomenon, however, is only visible for disturbances in the dynamics (first row of Figure 8).

For action disturbances, the average normalised return is not affected by increased training noise or testing noise, except at specific, high values where the average normalised return decreases dramatically, showing no obvious performance improvement. For noise added to observations, there

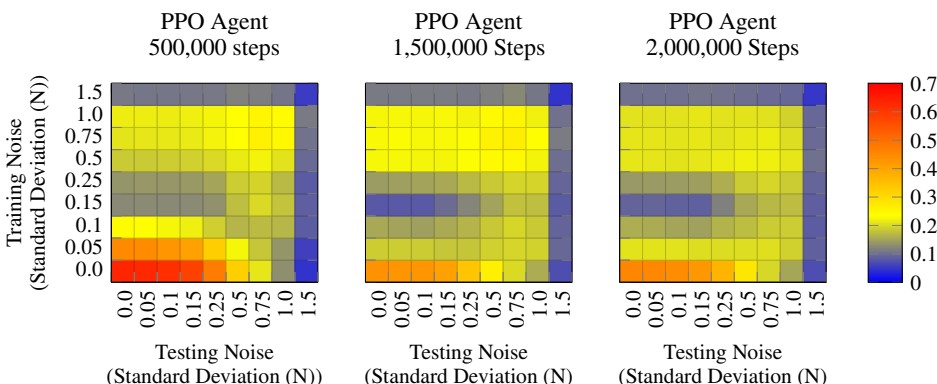

Figure 9: Heat maps of the average normalised return heat map of PPO trained (*y*-axes) and tested on (*x*-axes) varying levels of white noise on the dynamics for (left to right) 500,000, 1,500,000, and 2,000,000 steps.

is a sudden decrease to nearly zero average normalised return when noise is introduced during training. To make sure that the limited improvements in Figure 8 were not caused by a lack of training time in the more "difficult" environments with disturbances, we also trained the PPO agent an additional 500,000 and 1,000,000 steps. In Figure 9, the results from a shorter training (500,000 steps, leftmost plot), show that the performance is still lower than the best model. However, as the model is trained more, at 1,500,000 and 2,000,000 steps, the performance of the models trained with disturbances still worsens. We can also still see that high training noise leads to small performance improvement at high values of testing noise (i.e., going from the bottom right to the top right quadrants of the heat maps) although to levels that do not compare to the ideal performance. These small improvements also reduce as training continues. A possible explanation for this result is that the agent's models fail to simultaneously learn the behaviour of two systems, the cart-pole and the noise.

## 5   Limitations

The scope of our results is in part limited for reasons of brevity and timeliness. Yet, we recognise that the field of robust RL is vast and fast-paced and many other approaches are deserving of consideration. Results for three more approaches are included in the Supplementary Material. In Section 4, we leverage the traditional cart-pole stabilisation task to focus on robust RL agents and disturbance injection. However, we recognise that it is important to bring benchmarking efforts closer to deployable and interesting robotic systems, with higher degrees of freedom. The Supplementary Material also includes results for a quadrotor platform. In Subsection 4.3, we injected disturbances during training to study whether this would improve the agents' ability to cope with them during testing. This approach did not result in significant improvements. However, adversarial approaches and domain randomisation (see Supplementary Material) have shown approach-agnostic improvement and should be further explored.

## 6   Conclusions and Outlook

In this article, we presented results that provide insight into the robustness of reinforcement learning, in particular in the context of continuous control. One of our main findings for roboticists is that the RL agents are more susceptible to disturbances injected into dynamics and observations. On the other hand, all agents under test, both vanilla RL agents and robust ones, display some inherent robustness to action disturbances. In our experiments, robust approaches, although also capable of mitigating disturbances, did not consistently provide substantially better performance when compared to the traditional state-of-the-art RL. As the field of robust reinforcement learning develops, our results indicate that particular care should be dedicated to improving robustness to observation and dynamics disturbances. Nonetheless, building on traditional RL approaches that already demonstrate to generalise well against disturbances may be a promising path for robust robot control using reinforcement learning.

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

## Supplementary Material

**Tasks, Environments, and Disturbances Details**   The following results include two tasks: the cart-pole stabilisation task (as in the main body of the article) and a 2D quadrotor trajectory tracking task. To benchmark for robustness, [34] is utilised to inject disturbances in the environments for both training and evaluation. Concretely, we consider the standard observation and action noises—modelled as uniform or normal distributions, with varying scales. Disturbances in the dynamics are environment-specific; for cart-pole, it is a 2D random tapping force acting on the center-of-mass of the pole and for quadrotor, it is a random force acting on the center-of-mass of the UAV. Beyond disturbances, another important class of nonidealities is parameter mismatch. The tunable parameters are also environment-specific; for the cart-pole, they consist of the pole length, pole mass, and cart mass; for the quadrotor, they include the drone mass and moment of inertia.

**Additional Agents**   Beyond PPO, SAC, RARL, and RAP, the results in the following also include:

- **WCPG [29]**. A formulation in robust RL that optimises for worst-case performance. It models the full distribution of returns as in distributional RL and uses a risk-averse measure as its optimisation objective. In WCPG, the return distribution is parametrised as Gaussians $Z^\pi(s, a) \sim \mathcal{N}(Q^\pi(s, a), \Upsilon^\pi(s, a))$, where the mean and variance are both value functions to be learned with TD methods. With this parametrisation, the conditional value at risk (CVaR) of the returns can be computed in closed form, which captures the average of low-percentile returns as a surrogate to worst-case performance. Given a percentile threshold $\alpha$, the actor then uses CVaR of the returns $\Gamma^\pi(s, a, \alpha)$ in its loss gradient. We implement WCPG on top of a SAC agent with the new update rules for the return distribution variance and actor.

$$\Gamma^\pi(s, a, \alpha) = \text{CVaR}_\alpha = Q^\pi(s, a) - (\phi(\alpha)/\Phi(\alpha))\sqrt{\Upsilon^\pi(s, a)}$$
$$\phi(\cdot), \ \Phi(\cdot) \ - \ \text{PDF and CDF of standard normal distribution}$$
$$\nabla_\theta J_\alpha = \mathbb{E}_\pi[\nabla_\theta \log \pi_\theta(a|s)\Gamma^\pi(s, a, \alpha)]$$

- **RAAC [32]**. Similar to WCPG, this approach optimises for the CVaR of returns but parametrises the return distribution as value functions conditioned on the percentile $Z^\pi(s, a; \tau)$. These value functions can also be learned via TD methods and be used to approximate the returns' CVaR. The original work focuses on an offline RL setting but we adapt the online learning version called RAAC for comparison. Again, our implementation is based on a SAC agent with the additional update rules for the value function and actor.

$$\text{CVaR}_\alpha(Z^{\pi_\theta}(s, a; \tau)) = \frac{1}{\alpha} \int_0^\alpha Z^{\pi_\theta}(s, a; \tau)d\tau$$

$$\approx \frac{1}{\alpha K} \sum_{k=1}^{K} Z^{\pi_\theta}(s, a; \tau_k), \quad \tau_k \sim \mathcal{U}(0, \alpha)$$

- **Domain Randomisation (DR) [23, 31, 4]**. DR can be built on top of any standard RL algorithm (e.g., PPO). The key idea is to generate a set of training environments with randomised system parameters, forcing the agent to generalise over these environment variants and possibly even to environments out of the training distribution. Typically, DR requires careful selection of system parameters and tuning of the randomisation ranges.

**Additional Results**   The following additional results use the root-mean-squared error (RMSE) as their performance metric. RMSE is a common measure to compare controller performance in robotics given a goal state or reference trajectory (i.e., $x^{goal}$ in Equation (1)). The learning curves are shown in Figure 10 and 11.

In the cart-pole task from Figure 10, we observe that most robust approaches show increased performance and converge to close-to-zero RMSE except for RAAC, which exhibits large oscillations in its learning curve and does not stabilise in the given training time as the other agents.

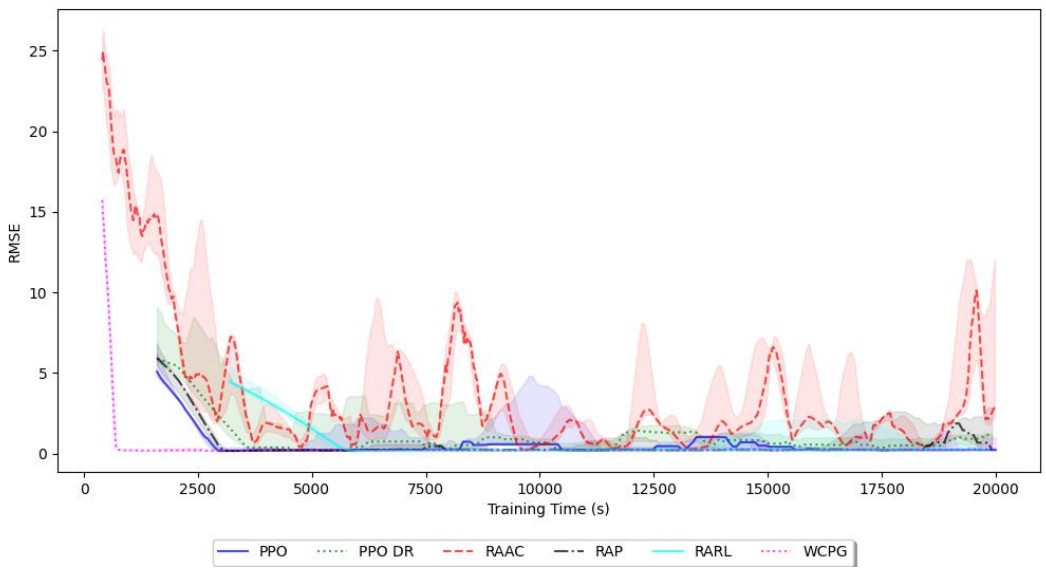

Figure 10: Learning curves of the RL agents and robust baselines on the cart-pole stabilisation task.

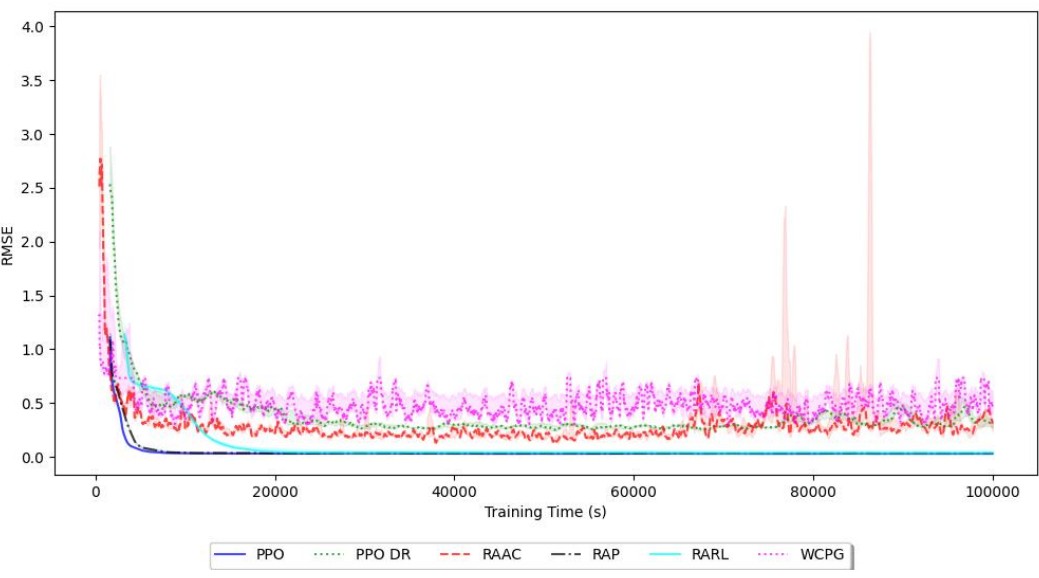

Figure 11: Learning curves of the RL agents and robust baselines on the quadrotor tracking task.

WCPG has a noticeably faster convergence rate, followed by PPO, RAP, and RARL. PPO and RAP have similar learning trends, while RARL requires more data to converge. This is possibly motivated by the fact that RAP is trained with action noise, but RARL is trained with an external tap force on the pole's centre-of-mass which intuitively poses a harder challenge to stabilise. PPO with domain randomisation (PPO DR) experiences an initial slow learning phase with high variance in RMSE, then a more stable performance trend till the end of training, which can be attributed to the need for sufficient exploration in the randomised training environments.

In the quadrotor task from Figure 11, again PPO and the two adversarial RL methods (RAP, RARL) converge to low RMSE and show similar learning efficiency trends as in the cart-pole task. The two

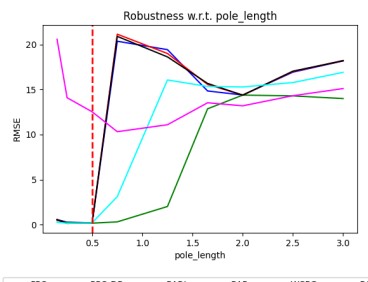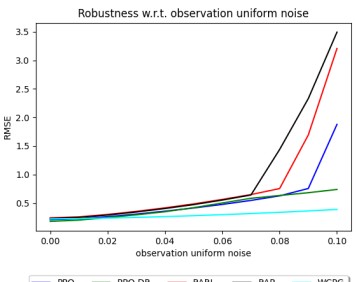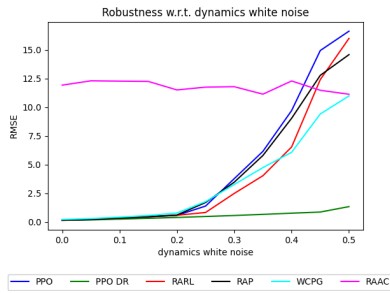

Figure 12: Robustness performance benchmark with one varying disturbance in cart-pole: *Left*: pole length (the vertical dash line represents default pole length in the training environment). *Mid*: observation noise (RAAC is not plotted due to a poor performance curve lying out of range from the others). *Right*: external tap force on pole center-of-mass.

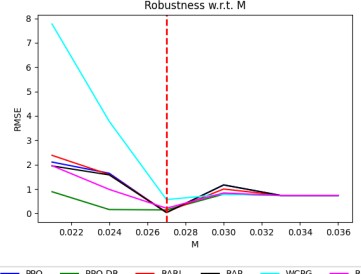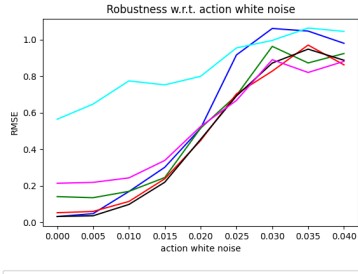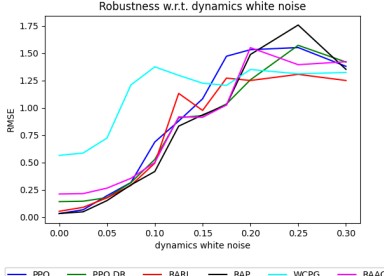

Figure 13: Robustness performance benchmark with one varying disturbance in quadrotor: *Left*: drone mass (the vertical dash line represents default drone mass in the training environment). *Mid*: action noise. *Right*: external 2D force on drone.

distributional RL-based methods (WCPG, RAAC) underperform in RMSE compared to PPO and the adversarial methods. Additionally, their learning curves show high-frequency oscillations. This effect might be due to several factors: (i) reliance on noisy sampling-based gradients for learning; (ii) learning value functions that capture the full distribution instead of the mean of returns, leading to requiring more samples.

WCPG does not exhibit the same behaviour in the cart-pole stabilisation task, possibly because of the lower task difficulty and rapid convergence to the optimal policy. WCPG also has larger oscillation magnitudes than RAAC due to the difference in training setup. Specifically, RAAC uses a fixed alpha threshold to optimise the $\text{CVaR}_\alpha$ of returns while WCPG uniformly samples a new alpha value from a predefined range for each training episode. Lastly, PPO DR also shows a slow learning trend at the initial stages, but it has a stagnant learning curve afterwards and fails to converge to low RMSE like PPO, indicating possible local minima. This could suggest difficulties in tuning domain randomisation parameters in more complex tasks.

These results shed light on features of the learning process among the robust approaches. To look at their final test performances against disturbances, we also benchmark them with different classes of disturbances and for each class, we evaluate the agents given different magnitudes of the disturbances.

Figure 12 shows the baseline performances in cart-pole for three kinds of disturbances: (i) varying pole lengths; (ii) observation noise from a uniform distribution of varying scales; and (iii) external 2D tap force on pole's center-of-mass from a normal distribution of varying scales. We observe that PPO DR has the best overall performance across most of these disturbances, proving its efficacy in learning robust policy despite the simplicity of the method. The two adversarial RL baselines (RARL, RAP) have similar performance trends to PPO with these disturbances, except RARL which shows a slight advantage with the external force after scale 0.3. This is to be expected since RARL is formulated to train against adversarial external forces.

Similarly, Figure 12 also suggests that RAP is more robust to action noise, given that it is formulated to train against it. These two results may indicate the limited generalisability of adversarial RL methods to other types of disturbances. Alternatively, it may also suggest training adversarially against a hybrid set of disturbances to attain broader generalisability. Unlike the adversarial methods that require predefined disturbances in the training environment, the distributional RL methods (WCPG, RAAC) optimise the CVaR of the returns without prior knowledge of any disturbance. WCPG shows comparable robustness to the adversarial methods and better performance in certain regions such as small variation of pole length or high perturbation of external force. Noticeably, WCPG is even outperforming PPO DR when facing observation noise.

RAAC is seemingly non-robust against the three tested disturbances, except in the left plot where RAAC shows better RMSE than some other agents when pole length variations are large. RAAC's underperforming compared to WCPG can be the result of using more complex parametrisation for the distributed value function as well as using a fixed CVaR threshold throughout training.

Moving on to the harder quadrotor tracking task in Figure 13, PPO DR no longer shows an obvious advantage over the other approaches across the tested disturbances, except in the low quadrotor mass region in the left plot. By contrast, PPO, RAP, and RARL again have similar performance trends, specifically RAP and RARL outperform PPO in their respective disturbance domains.

Another difference is the robustness performance of WCPG and RAAC. Given the harder task, RAAC starts to outperform WCPG against all disturbances and it is comparable to the adversarial RL baselines. WCPG falls short in this harder task, placing behind other approaches and only closing the gap with large enough disturbances. This is possibly due to the benefit of more representation power, since WCPG parametrises the return distribution only as Gaussian while RAAC parametrises it as quantiles that ideally can capture broader distributions, making RAAC more robust to complex changes to the environment.

To further investigate how the trained robust agents perform in the presence of hybrid disturbances, in the following, we show the benchmark results by evaluating the agents for two disturbances.

Figure 14 shows a set of performance heatmaps for each baseline with varying pole lengths and pole masses in the cart-pole task. By looking at the scale and distribution of the heatmap values, we conclude that PPO DR has the overall most robust performance against system parameters change.

Besides, WCPG shows better robustness to system parameter change compared to other baselines. PPO, RARL, and RAP have similar heatmap patterns, indicating similar performances in terms of robustness to system parameters. Due to suboptimal learning (Figure 10) RAAC's heatmap shows higher RMSE and a noisier distribution compared to the other approaches.

We also observe that most agents except RAAC have some similar patterns: (i) scaling both system parameters results in lower RMSE than scaling one parameter alone, as seen from the bottom left and upper right corners in the heatmaps; (ii) pole length has higher sensitivity than pole mass as seen from the large RMSE change column-wise than row-wise, and (iii) lower pole lengths seem easier for control than higher ones, with most of the low RMSE values concentrated in the bottom region.

For the quadrotor trajectory tracking task, Figure 15 shows the performance heatmaps with varying quadrotor mass and action noise from a normal distribution of varying scales. All of the approaches have similar heatmap patterns but differ in the RMSE scales. Specifically, PPO, PPO DR, RARL, RAP, and RAAC have roughly equal RMSE scales while WCPG has higher RMSE values, indicating WCPG's lesser ability to generalise in the face of a mixture of system parameter changes and action noise in the quadrotor more complex task.

Some common patterns are again observed from the heatmaps: (i) smaller drone masses are harder to control than larger ones and incur higher RMSE regardless of action noise as seen in the bottom region; (ii) the agents perform better given larger action noise, which is especially visible in the bottom rows of WCPG.

**Ablations: Domain randomisation range**    To provide further insights into robustness benchmarking and guidance for the design of future robust RL agents, we also perform a series of ablations w.r.t. key parameters. To start, PPO DR has shown robustness against various disturbances in the cart-pole task but its performance is not as good in the quadrotor task, and we argued that the tuning

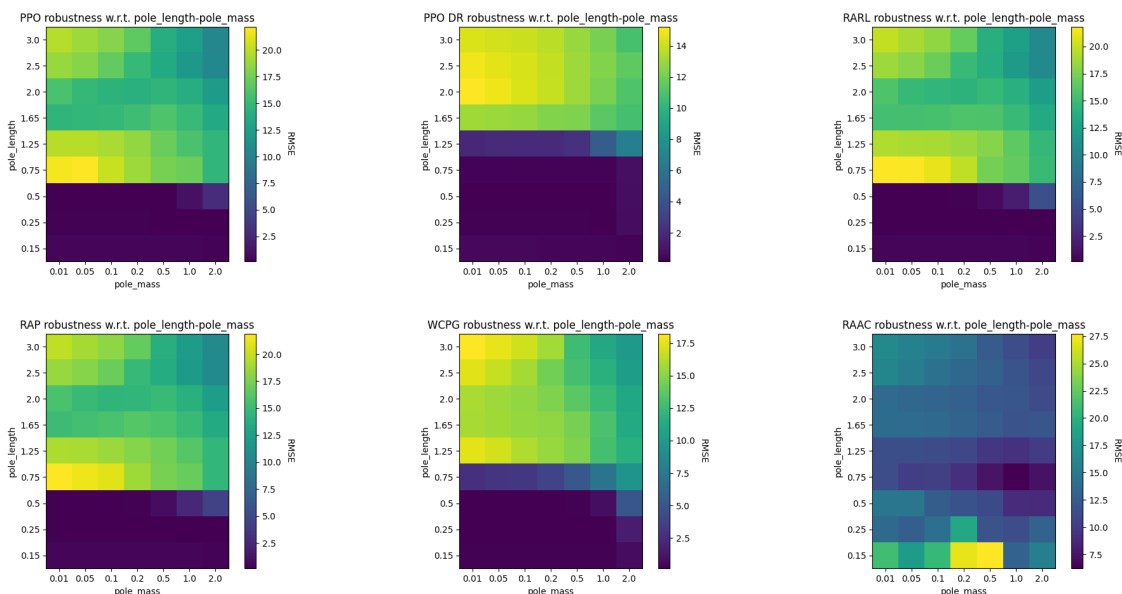

Figure 14: Robustness performance heatmaps with two varying disturbances (pole length and pole mass with default values 0.5 and 0.1) in cart-pole for each agent.

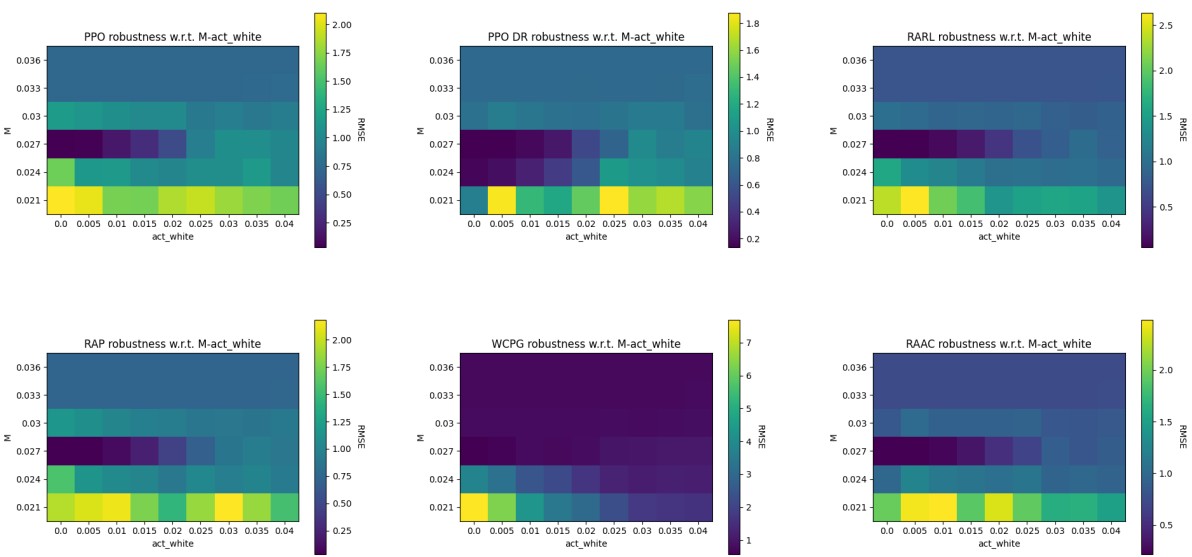

Figure 15: Robustness performance heatmaps with two varying disturbances (drone mass and action noise with default values 0.027 and 0) in quadrotor for each agent.

Table 1: Domain randomisation ablation in the cart-pole task.

| Tests | PPO | PPO DR (varying training DR range) | | |
|---|---|---|---|---|
| | | low | mid | high |
| default | 0.179 ± 0.092 | 0.148 ± 0.074 | 0.16 ± 0.087 | 0.276 ± 0.043 |
| params x1.5 | 17.272 ± 2.883 | 11.379 ± 3.166 | 0.325 ± 0.07 | 0.181 ± 0.047 |
| params x8 | 6.876 ± 0.639 | 6.133 ± 0.787 | 5.058 ± 1.089 | 3.638 ± 0.896 |
| obs + act noise | 2.531 ± 5.961 | 0.501 ± 0.108 | 0.629 ± 0.1 | 0.347 ± 0.038 |
| tap force | 20.582 ± 7.703 | 9.987 ± 6.934 | 2.349 ± 1.954 | 1.511 ± 0.566 |

of domain randomisation parameters can be a crucial factor. To investigate, we benchmark PPO DR with different levels of randomisation ranges: low, mid, and high:

- **default**: pole length - 0.5, pole mass - 0.1, cart mass - 1.0
- **low**: pole length $\sim \mathcal{U}(0.3, 0.7)$ pole mass $\sim \mathcal{U}(0.05, 0.2)$ cart mass $\sim \mathcal{U}(0.7, 1.3)$
- **mid**: pole length $\sim \mathcal{U}(0.1, 1.0)$ pole mass $\sim \mathcal{U}(0.01, 0.5)$ cart mass $\sim \mathcal{U}(0.1, 2.0)$
- **high**: pole length $\sim \mathcal{U}(0.1, 3.0)$ pole mass $\sim \mathcal{U}(0.01, 2)$ cart mass $\sim \mathcal{U}(0.1, 4.0)$

The default parameters are from the nominal cart-pole environment [34] and correspond to training a simple PPO agent without domain randomisation. The mid-level parameters are used to train PPO DR and benchmark with other baselines in the previous section. The low-level and high-level parameters are sampled from a smaller and a larger randomisation range respectively.

Table 1 shows the ablation results of varying levels of domain randomisation against different types of evaluation environments. Although in the nominal cart-pole environment PPO DR only narrowly improves over PPO, we observe that domain randomisation can effectively counter the presence of disturbances against which PPO falls short.

In the scenarios where the system parameters are scaled by 1.5 times and 8 times, PPO DR has better performance when the randomisation range also scales with the deviation of the test evaluation environment to the nominal training environment. PPO DR with high-level randomisation turns out to be more effective than the one with mid-level randomisation in the 1.5 times scaled cart-pole, although the latter is trained in environments closer to the test environment.

This might suggest a larger randomisation range does not always sacrifice individual performance in exchange for better overall performance. A potential reason could be that the larger randomisation not only aids in robustness to disturbance but also induces better training regularisation. Domain randomisation also generalises to other forms of disturbances such as noise and external force.

From the ablations, a larger randomisation range seems consistently useful to cope with different disturbances. Hence, the tuning of domain randomisation remains largely a factor of availability of training resources, prior knowledge, and trial-and-error.

**Ablations: Adversary disturbance scale** One of the classes of benchmarked approaches are the adversarial RL methods. They showed robustness but limited to the classes of disturbances that they have been trained on. This naturally suggests that combining multiple disturbances during adversarial training could achieve broader robustness.

However, a crucial issue in adversarial RL is tuning the adversaries' strength. It has been pointed out in previous work [20] that an overpowered adversary can easily undermine the learning of the main policy. Yet, a weak adversary is less effective in encouraging robust behaviours.

In Table 2, we show ablations for RARL and RAP with varying adversarial disturbance scales tested in differently disturbed environments. As we expected, RARL and RAP perform better with external tap forces (rows 4,5) and action noises (rows 2,3), respectively.

As we also noted in the main body of the article, it is worth pointing out that the baseline PPO scores similarly to RAP in both the default nominal environment and the disturbed environment with action noises. This may indicate PPO is already equipped with intrinsic robustness to action disturbances, such robustness is even competitive to RAP when the disturbance is small (row 2).

On the other hand, although RARL displays poor performance in both the nominal and action disturbed environments, it generalises better than the others when system parameters are changed.

Table 2: Adversary disturbance scale ablation in the cart-pole task.

| Tests | PPO | RARL (varying scale) | | | | RAP (varying scale) | | | |
|---|---|---|---|---|---|---|---|---|---|
| | | 0.01 | 0.1 | 0.5 | 1.0 | 0.01 | 0.1 | 0.5 | 1.0 |
| default | 0.215 | 0.526 | 4.744 | 2.978 | 6.845 | 0.225 | 0.211 | 0.244 | 0.226 |
| act noise $\sim \mathcal{N}(0,1)$ | 0.197 | 0.247 | 3.587 | 2.107 | 7.092 | 0.226 | 0.211 | 0.478 | 0.231 |
| act noise $\sim \mathcal{N}(0,3)$ | 0.264 | 0.337 | 5.027 | 2.279 | 6.463 | 0.798 | 0.256 | 1.582 | 0.269 |
| tap force $\sim \mathcal{U}(-1,1)$ | 20.142 | 16.479 | 21.733 | 8.075 | 9.513 | 16.386 | 19.607 | 19.754 | 19.461 |
| tap force $\sim \mathcal{U}(-3,3)$ | 25.542 | 27.377 | 26.929 | 11.461 | 12.738 | 25.134 | 26.398 | 28.189 | 27.024 |
| params x1.5 | 17.617 | 17.861 | 18.018 | 5.787 | 6.331 | 16.870 | 16.932 | 17.555 | 17.415 |

Table 3: CVaR$_\alpha$ threshold ablation in the cart-pole task.

| Tests | PPO | WCPG (varying $\alpha$) | | | | RAAC (varying $\alpha$) | | | |
|---|---|---|---|---|---|---|---|---|---|
| | | 0.1 | 0.3 | 0.5 | 1.0 | 0.1 | 0.3 | 0.5 | 1.0 |
| default | 0.202 | 0.585 | 0.279 | 0.219 | 0.238 | 17.143 | 6.188 | 12.352 | 21.721 |
| params x1.5 | 17.656 | 5.659 | 10.305 | 6.703 | 6.918 | 11.957 | 4.678 | 7.03 | 11.311 |
| obs + act noise | 2.135 | 0.649 | 0.369 | 0.328 | 0.319 | 15.928 | 5.956 | 11.645 | 22.757 |
| tap force | 20.476 | 8.276 | 15.608 | 10.34 | 10.157 | 18.704 | 8.538 | 13.766 | 22.35 |

**Ablations: Risk measure threshold**  The final class of robust methods we considered are the distributional RL ones. These have the advantage that no prior information on the type of disturbance is needed. In our benchmark, both WCPG and RAAC use CVaR$_\alpha$ of the returns as the risk measure, and an important design decision is the selection of the risk measure threshold $\alpha$. Specifically, the threshold implicitly controls the weightings of learning data and affects how conservative each policy update is, which is eventually reflected in the final policy's robustness.

Table 3 shows WCPG and RAAC's performance for different risk measure thresholds in training and multiple testing cart-pole environments. We should note that the CVaR threshold $\alpha = 1$ is equivalent to the mean of the return distribution, thus RAAC with $\alpha = 1$ is similar to a traditional actor-critic method. WCPG outperforms the other agents under observation-action noise by a large margin, but only slightly wins over RAAC under the presence of an external force. Interestingly, the WCPG variants show similar performance in noisy environments as in the nominal environment.

However, we also observe that there is not a unique threshold that achieves the best performance over all the tested environments. RAAC only has better performance for system parameter changes. Nonetheless, there seems to be a consistent trend in which RAAC with threshold $\alpha = 0.3$ is the best variant compared to other thresholds over the different disturbances.

