# OpenReview forum: "Characterising the Robustness of Reinforcement Learning for Continuous Control using Disturbance Injection"
_NeurIPS.cc/2022/Workshop/TEA — TEA_

### Official Review · Reviewer_4n7g · 2022-10-15
**An empirical study of RL-based policies on robustness to different types of disturbance.**

**Rating:** 5
**Confidence:** 4

**Review:**

This paper proposes to characterize the robustness of reinforcement learning policies by injecting disturbance in observations, actions, and dynamics. Quantification is essential for verifying autonomous systems in the real world. However, the approach is not well described, and the analysis is limited.

My major concerns are
1. The comparison lacks essential information. For the robust reinforcement learning (RL) approaches, the authors do not explicitly mention the bound of the imaginary adversarial agent(s). The slight improvement of the robust approaches can be due to the fact the adversary's capability is much weaker than the uncertainties considered in the evaluation settings.
2. Authors claim that RL policies are most robust in action disturbances. However, the action disturbance considered can be viewed as a particular type of dynamic disturbance. It will be good for authors to quantify the effect of the proposed disturbance on the state variables.
3. Authors provide training with "random" disturbances, also known as the domain randomization approach. However, their results show that training RL with higher noises does not perform better even under the same noise level. This is counter-intuitive and may indicate that the training is not set up correctly. Additionally, RAP considers the worst realization of the uncertainty, so adding noises in RAP training is weird.

Minor comments:
1. There are no error bars or shaded confidence intervals in the figures, which makes it challenging to analyze the effect of random trials.
2. The normalization in Fig. 3-9 is unclear. What is the baseline?

---

### Official Review · Reviewer_BgE2 · 2022-10-15
**Useful comparison of state-of-the-art RL methods with injected disturbances; however, results are in simple settings**

**Rating:** 6
**Confidence:** 4

**Review:**

**Summary:** The authors test the robustness and safety of a set of RL methods on the standard cart-pole problem by injecting various kinds of noise to the observation, action, and dynamics. The authors show that, on cart-pole, the robust RL agents do not perform significantly better than the standard RL agents and that all the RL agents are more robust to disturbances injected through actions as compared with observation/dynamics injection.

**Clarity / Quality:**
The methods tested and sources of disturbance are well-explained. I would appreciate the presentation of the algorithms for the RL methods tested (PPO, SAC, RARL, RAP, WCPG, RAAC, DR). Additionally, the authors should include the variance in Figures 1,3,4,5,6,7,12,13 to help the reader understand the stability/reliability of the method in the presence of the disturbance.

**Originality:** The authors do not contribute any theoretical results or algorithms. However, I’m not aware of a work which has such systematic comparions of RL methods for the wide variety of disturbances types and sources considered.

**Significance:** The results are interesting and of value to the RL community. My main concern with the significance is that the experiments are only run on the cart-pole problem or 2D quad rotor trajectory tracking problem. Although the authors do demonstrate interesting results, it’s not clear that these results would be consistent for more challenging RL problems with i.e. higher dimensional state/action spaces.

**Issues:**
- The authors find that robust RL agents do not perform significantly better than standard RL agents in the scenarios considered. This is certainly an interesting finding and motivating for future work. However, it’s not clear that this trend would hold up on challenging RL problems with higher dimensional state/action spaces
- The authors mention that they hope their work can provide a basis for further research. I hope the authors would make their code publicly available so that the community directly leverage their comparisons.
- L213-L214 "The evaluation/test performance with higher levels of noise is generally still better when training with low levels of noise…” Is there an error here? It seems like, based on Figure 8, that less noise in training is better across the board.

---

### Official Review · Reviewer_wSZS · 2022-10-17
**Study on RL Robustness**

**Rating:** 7
**Confidence:** 4

**Review:**

This paper studies the robustness of controllers trained using RL to disturbances. The key insight presented by the paper is that the sensitivity of the RL controller is very high for disturbances to the observations and dynamics and low for disturbances to actions.

Understanding the robustness of RL policies to disturbances and distributions shifts is an important undertaking, particularly since they are touted as the next generation of control synthesis strategies for embodied AI. I am leaning towards an accept for this workshop paper, but I believe a discussion section should be included. It is unclear to me what the scope of these findings are, how or where can we apply them, and why do we observe higher robustness to disturbances in the actions compared to observations and dynamics.

---

### Decision · Program_Chairs · 2022-10-21

**Decision:**

Accept

**Comment:**

This paper is a very interesting first step on systematically analyzing the robustness of RL policies under different types of disturbances. It is an important problem towards safe application of RL in embodied AI. It could make a good contribution to the workshop. Meanwhile, the reviewers pointed out a few limitations of the current experimental settings. While it might not be possible to address all these concerns at this point, please carefully consider the reviewers' comments in the final version. In particular, it would be helpful to discuss how the findings in these relative simple experiments will be helpful for future exploration (e.g., challenging RL problems with high-dimensional state and action spaces).